# Evidence on Child Nutrition Recommendations and Challenges in Crisis Settings: A Scoping Review

**DOI:** 10.3390/ijerph18126637

**Published:** 2021-06-20

**Authors:** Aniqa Islam Marshall, Gideon Lasco, Mathudara Phaiyarom, Nattanicha Pangkariya, Phetdavanh Leuangvilay, Pigunkaew Sinam, Rapeepong Suphanchaimat, Sataporn Julchoo, Watinee Kunpeuk, Yunting Zhang

**Affiliations:** 1International Health Policy Program, Ministry of Public Health, Nonthaburi 1100, Thailand; aniqa@ihpp.thaigov.net (A.I.M.); nattanicha@ihpp.thaigov.net (N.P.); pigunkaew@ihpp.thaigov.net (P.S.); rapeepong@ihpp.thaigov.net (R.S.); sataporn@ihpp.thaigov.net (S.J.); watinee@ihpp.thaigov.net (W.K.); 2Department of Anthropology, University of the Philippines Diliman, Quezon City 1107, Philippines; pdlasco@upd.edu.ph; 3Development Studies Program, Ateneo de Manila University, Diliman, Quezon City 1106, Philippines; 4Equity Initiative, Bangkok 10110, Thailand; Leuangvilaypdv@gmail.com (P.L.); edwinazhang@hotmail.com (Y.Z.); 5Division of Epidemiology, Department of Disease Control, Ministry of Public Health, Nonthaburi 11000, Thailand; 6Child Health Advocacy Institute, Shanghai Children’s Medical Center, Shanghai Jiao Tong University School of Medicine, Shanghai 200127, China

**Keywords:** child nutrition, crisis settings, emergency, public health responses

## Abstract

Adequate child nutrition is critical to child development, yet child malnutrition is prevalent in crisis settings. However, the intersection of malnutrition and disasters is sparse. This study reviews existing evidence on nutrition responses and outcomes for infants and young children during times of crisis. The scoping review was conducted via two approaches: a systematic search and a purposive search. For the systematic search, two key online databases, PubMed and Science Direct, were utilized. In total, data from 32 studies were extracted and included in the data extraction form. Additionally, seven guidelines and policy documents were included, based on relevance to this study. Overall, the existing evidence demonstrates the negative impacts of crises on nutritional status, diet intake, anthropometric failure, and long-term child development. On the other hand, crisis-related interventions positively affected nutrition-related knowledge and practices. Further studies should be carried out to explore the sustainability of the interventions and the success of existing guidelines. Since this study focuses only on nutrition among children under three, further studies should likewise consider an extended age range from three to five years.

## 1. Background

Early child development—defined as the years from birth to three—is a crucial period for ensuring an individual’s physical and mental health and is positively associated with higher income and productivity later in life [1]. Thus, it has emerged as a global public health concern, as evidenced by several international initiatives in recent years [2,3,4]. In low-income and middle-income countries, where crises are protracted, children’s development is put at risk [4]. Nutrition is among the most important risk factors for early child development: latest estimates suggest that about 250 million (43%) children under five in low and middle-income countries were at risk of not achieving their full development potential due to stunting and extreme poverty [5].

Despite existing nutrition plans in most southeast Asian countries, such as the Philippine Plan of Action for Nutrition 2017–2022 and Thailand’s 2nd National Policy for Reproductive Health and Development, the prevalence of malnutrition remains unacceptable. A World Health Organization (WHO) bulletin [6] reported that more than a quarter (27.9%) of children under five are stunted, which affects their physical health, cognitive development, and socio-economic status. Inadequate dietary intake and diseases mainly contribute to malnutrition, which is the principal target of nutrition programs [6].

One significant but overlooked determinant of child nutrition, particularly in low- and middle-income countries, involves crisis situations, which encompass a wide range of calamitous events, from human-made disasters (humanitarian crisis, such as civil war or forced displacement) to natural disasters (such as typhoons and earthquakes) and also economic or financial crises [7]. Within these countries, poor and marginalized communities are disproportionately impacted by both disasters and malnutrition [8,9,10,11], but little is known about the intersection of these two domains. This study aims to fill this gap by reviewing existing global evidence of nutrition response for early child development during crises. In doing so, it can address the issue of nutrition in the context of disaster response, which plays an essential role in public health considerations in the region.

## 2. Methods

We followed Arksey and O’Malley’s methodological framework for conducting a scoping study [12]. The five stages of the method include: (1) identifying the research question, (2) identifying relevant studies, (3) selecting the studies, (4) charting the information and (5) summarizing the results.

### 2.1. Research Question

Based on the recommendations of Levac et al. [13], we developed our research question by taking into consideration the concept, target population and health outcomes of interest. The aim of this study is to identify current evidence, policies and responses during crisis situations for nutrition in children under three years old among low- and middle-income countries. Our primary research question was “What responses have been implemented to mitigate the nutritional challenges during crisis situations for children under three years in low- and middle-income countries”. We define the term “crisis” based on the WHO definition [14] as “an event or period of time that is perceived to be difficult”. We grouped crisis situations into three categories: (1) economic crisis, (2) natural disasters and (3) humanitarian crisis.

To answer our primary research question, we identified three subsidiary questions: (1) What is the impact of the crisis on child nutrition? (2) What are the impacts of child nutrition interventions during crises and the critical gaps and challenges of implementation? (3) What guidelines exist for addressing child nutrition during crisis situations?

### 2.2. Identifying Studies

The review was conducted between 18 June 2020 and 31 July 2020. The literature was explored through two approaches: a systematic search and purposive search. For the systematic search, two key online databases, PubMed and Science Direct, were utilized. The search terms applied in the databases were “Nutrition” AND “Child” AND (“Crisis” OR “Disaster”). Additional articles were also identified using snowball methods including examining the reference lists of pertinent articles [15]. For the purposive search, policies and guidelines relevant to child nutrition in crisis settings were manually searched and retrieved from websites of inter-governmental and non-governmental organizations such as the WHO, United Nations Children’s Fund (UNICEF) and Infant Feeding in Emergencies (IFE) Core Group.

Inclusion and exclusion criteria were applied to ensure article relevance to the study aim. Studies were included if they (1) provided evidence on the issue of nutrition; (2) included or were applicable to children under the age of three; and (3) were in the context of a natural or human-made crisis in developing countries. As mentioned previously, we defined the term “crisis”, based on the WHO definition, as an event or period of time that is perceived to be difficult [14] and grouped crisis situations into three categories: (1) economic crises, (2) natural disasters and (3) humanitarian crises. Studies that (1) focused on high-income countries, (2) addressed non-nutrition-related impact and interventions, (3) were in non-crisis situations, or (4) were irrelevant to children under three years were excluded. Articles published in languages other than English were also excluded due to limited capacity for language translation.

All articles obtained from the database searches were screened for duplication. Two authors (AM, MP) independently reviewed the titles and abstracts of unique articles based on the inclusion and exclusion criteria. Disagreements between the reviewers were resolved by discussion and consensus with a third party (RS). A full text review was conducted of the selected articles. Data were independently extracted by AM, MP, PS, SJ, and WK and put into a data extraction form. Prior to compiling all data into a single master sheet, all authors met to review all selections for agreement. Discrepancies were resolved by consensus, and if consensus could not be reached, a third party (RS) was invited.

### 2.3. Charting the Information and Summarizing the Results

All results were charted into an Excel template, with the following information collected: (1) Title, (2) author, (3) country or location, (4) type of crisis, (5) aim of the article, (6) type of study, (7) type and details of intervention, (8) type of study assessment, and (9) key results of the study.

## 3. Results

The literature search identified 144 articles, 105 from PubMed and 39 from Science Direct, of which 13 were excluded on the basis of duplication. Additionally, 11 articles were obtained through screening of reference lists of included articles. Two independent reviewers (AM and MP) initially screened 142 articles by title and abstract, excluding 98 articles based on the inclusion and exclusion criteria. A total of 44 articles were assessed in full, of which 12 were excluded and 32 included and extracted and put into the data extraction form. Figure 1 presents the full screening process. There were 22 papers that evaluated a nutrition intervention and the impact of nutrition during crisis situations, while 18 papers looked at the impact of crisis on nutrition or reviewed the impact of interventions during crisis. The papers were categorized into three groups of crisis situations: we identified 10 papers on economic or political crises, 11 on humanitarian crises, and 11 on natural disasters. Additionally, 12 guideline and policy documents were identified from the web search, of which seven were included based on relevance to this study. Overall, there were ten cross-sectional studies, six cohort studies, four secondary data analysis studies, four document review studies, two primary qualitative studies, two cluster randomized trials, two pre-post evaluation studies, one quasi-experiment study, and one non-randomized trial study (see Table 1).

### 3.1. Impact of Crisis on Nutrition

There were 15 articles which evaluated the impact of crisis on child nutrition. Thirteen were country specific—six Asian, six African, one Caribbean and one Eastern European—while two studies were regional/global studies (see Appendix A).

Studies applied several methodologies to assess child nutrition in crisis situations. Anthropometric measurements using survey data or interviews were the most common method to identify stunting, underweight, and wasting among children [16,17,18,19,20,21,22,23,24,25]. Studies [17,19,23,24,26,27,28] also assessed prevalence of anemia or malnutrition through assessments of blood samples.

The review identified seven papers that discussed the impact of economic crises on nutrition [17,25,29,30,31,32,33]: six during natural disasters [18,34,35,36,37,38], including cyclones, earthquakes, flooding and drought, and two during humanitarian crises [27,39]. Studies found that the prevalence of child morbidity and mortality was high in crisis situations, with worsened child diet and nutritional status, including higher prevalence of malnutrition and lower birth rates. Nidzvetska et al. (2017) [39] reported crisis situations leading to the displacement of children had an effect on inadequate dietary intake for children due to ration cuts and opting for alternative foods in cereals that were deemed cheaper. Similarly, Mulder-Sibanda [31] reported dramatic effects on child nutrition leading to high prevalence of stunting and underweight during a political crisis. Rajmil et al. [33] reported that an economic crisis was related to worsening nutrition habits. In addition, there were around 28,000–50,000 infant deaths in sub-Saharan Africa in 2009 resulting from crises. In Congo, the prevalence of low birth weight and morbidity among children under one year old was found to be high as a result of nutritional changes during economic crises [29]. Choudhury et al. [18] reported that the proportion of severally malnourished children increased significantly from 5% to 11% following floods; similarly, Dong et al. [38] reported a significant increase in malnourished children following earthquakes. However, interestingly, a review of the impact of drought on child undernutrition found that, though there are higher levels of undernutrition during droughts, the crisis itself is not the cause, but rather, only a trigger for already vulnerable settings [35].

Anemia was associated with economic and natural disasters, resulting in overall increased prevalence among children under 24 months, especially in already poor rural areas [17,37]. Several studies also reported on the anthropometric failures of children in crisis situations [17,19,25,29,31,32,34,36,37,38]. Mulder-Sibanda [31] reported that, during periods of economic prosperity, the prevalence of stunting, underweight, and wasting decreases significantly, and political and economic crises slow down improvements in stunting and underweight prevalence, while doubling the prevalence of wasting. Three studies [25,29,36] found an increased prevalence of wasting: four [17,34,37,38] found an increased prevalence of stunting and two [19,32] found an increased prevalence of underweight as results of crisis. In Nigeria and Congo, economic and political crises resulted in a significant increase in wasting [25,29]. In China, natural disasters, specifically the Wenchuan earthquake, increased wasting prevalence from 1.3% to 4%, stunting prevalence from 6.6% to 10.8%, and underweight prevalence from 0% to 5.9% [37], while economic crises increased prevalence of stunting from 5.7% to 9.1% in children under 6 months old and from 6.7% to 12.5% in children 6–12 months old [17]. In Bangladesh, natural disasters increased the likelihood of stunting and being underweight by 8% and 13%, respectively; similar natural disasters in Nepal resulted in a high prevalence of stunting (43%), with increased odds of moderate and severe stunting with odds ratio of 0.42 and 0.59, respectively [34,38]. In Cameroon, severe economic crises were associated with increased prevalence of stunting (from 23% to 29%) and being underweight (from 16% to 22%) among children under three years old [32]. Natural disasters were reported to especially have long-term impacts on a child’s nutrition by disruptions in food supply or diarrheal illness caused by contaminated water [18,34,35,36,37]. Dong et al. [37] reported that children’s nutritional status did not recover following earthquakes, but instead continued to worsen. Ninno et al. [36] reported that children’s long-term growth was unable to recover after being affected by flooding; similarly, Ahsanuzzaman et al. [34] reported that cyclones hinder the development of a whole generation of children.

Major factors associated with the impact of a crisis on child nutritional status included a child’s sex [18,37], age [17,18,29,32,38] and maternal education [17,29,32]. Choudhury et al. [18] reported an increased risk of malnutrition among boys compared to girls, with the most affected age group being children less than 18 months old, while Dong et al. [37] reported a higher percentage of anemia among girls than boys (25% to 71.5% for girls and 43.6% to 60.9% for boys) following crisis. Gaire et al. [38] reported that children aged 6–11 months were more likely to be moderately stunted, while children aged 36 to 47 months were more likely to be severely stunted. Pongou et al. [32] found that economic crises have more of an effect on children older than 5 months old in comparison to newborns, which may be a result of breastfeeding in younger age groups.

### 3.2. Child Nutrition Interventions in Crisis Situations

In this scoping review, we identified 23 articles on various nutrition interventions implemented during crisis situations and their impact on child nutrition outcomes. Of these, two papers reviewed the overall impact of nutrition interventions [35,40]. Two were categorized as cash-for-nutrition programs [41,42], one as a food-for-work program [28], six as nutrition education interventions [16,24,26,41,43,44], one as a malnutrition screening study [41], two as nutrition policy interventions, and the majority (13) as supplementary feeding programs [18,19,20,21,22,23,24,27,28,36,44,45,46]. Additionally, four programs were combination interventions, one being cash-for-nutrition with nutrition education and malnutrition screening [41], one as food-for-work with supplementary feeding, and two as supplementary feeding with nutrition education [24,44].

Balhara et al. [40] reported that nutrition interventions decrease pediatric morbidity and mortality in humanitarian crisis situations. However, a systematic review on the effect of drought on child nutrition found that there was varying effectiveness of nutrition interventions during droughts, and interventions to address the critical levels of child undernutrition during droughts are lacking. The review [35] reported a decline in child underweight levels following interventions, as well as studied with limited evidence of mitigating impact of interventions.

The two cash-for-nutrition programs were evaluated for effectiveness of the interventions in humanitarian settings. Kurdi et al. [41] reported that the intervention in Yemen which provided monthly cash transfers of 10,000 Yemeni riyals (25% of the value of average monthly food spending), conditional on attendance at monthly nutritional training sessions led by locally recruited community health volunteers, was effective in improving women’s knowledge and practices on child nutrition and breastfeeding. In Somalia, a monthly unconditional cash transfer of USD 84 for 5 months increased monthly household expenditure by USD 29.60 and the household Food Consumption Score by 14.8 [43]; however, the intervention did not appear to reduce the risk of child malnutrition (unadjusted hazard ratio of 0.83) [42].

Of the six studies based on nutrition education programs, the majority (five) focused on breastfeeding outcomes [16,26,41,43,44]. According to MirMohamadaliIe et al. [43], training programs and distribution of materials on breastfeeding can help overcome barriers to breastfeeding in disaster situations. Six days of nutrition education with a focus on early initiation significantly increased the probability of breastfeeding initiation within the first hour of birth by 15.6%, exclusive breastfeeding for six months by 15.6%, and mothers’ knowledge of breastfeeding by 17.7% [41]. Similarly, an intervention consisting of door-to-door visits and training by community health workers found significantly longer exclusive breastfeeding and higher proportion of exclusive breastfeeding (92%) compared to 51% in areas without intervention [16]. Another 3-day nutrition training program primarily focused on addressing anemia in children found improvements in both knowledge and practice on nutrition related to anemia treatment, resulting in increased hemoglobin concentration by approximately 4 g/L and decreased anemia prevalence, from 14.3% to 7.8% [26]. Nutrition education programs were also found to have resulted in increased utilization of treated drinking water for children under two by 16% [41], higher infant weight at 12 months (mean weight of intervention group compared to non-intervention group = 8.42 vs. 7.95 kg) [16], and decreased child mortality [16]. Kurdi et al. [41] reported that the success of nutrition education programs was attributable to local community health volunteers assigned to conduct trainings, resulting in stronger trusting relationships with mothers and a decreased need for strict administrative oversight. These community health volunteers also successfully conducted malnutrition screenings in parallel to nutrition education programs to identify malnourished children and refer these children to health centers for treatment [41].

Supplementary feeding and food aid programs resulted in significant reductions in malnutrition and anthropometric failures in children. Targeted feeding programs in India reduced undernutrition in children from 66.7% to 59.6% following drought [44]. In Indonesia, an intervention providing meals, supplements, and snacks to children following a financial crisis resulted in significant changes from severe stunting to stunting in children under 5 years old [20]. In Sri Lanka, food aid consisting of high-energy biscuits for children with acute malnutrition decreased wasting from 18% to 9.6% and increased recovery rates from malnutrition [22]. Another food aid program found a lower prevalence of acute malnutrition and decrease in odds of child underweight (adjusted odds ratio of 0.40 compared to 0.82) following earthquakes [21]. Iron and vitamin supplements in Palestine decreased wasting from 6% to 1.4%, underweight prevalence from 10.9% to 3.8%, and anemia prevalence from 30.1% to 18.8% [24]. Vitamin A supplementation following flooding protected children from severe malnutrition in Bangladesh [18]. A supplementary feeding program containing micronutrient-fortified spreads successfully treated 67% of children from malnutrition in Guinea-Bissau [27]. At Saharawi refugee camps, a similar program to address anemia increased linear growth by 30% compared to children that did not receive the intervention, as well as decreased stunting and increased hemoglobin concentrations, resulting in reduction in anemia prevalence by nearly 90% [23]. Similarly, in China, a daily supplement containing proteins, vitamins and minerals decreased wasting from 3.5% to 1.7%, stunting by 9.9% to 50% and underweight prevalence, as well as reduced prevalence of anemia from 74.3% to 37.4% [23]. However, in Kenya, similar supplementation only caused a small improvement in iron status and no significant changes to hemoglobin levels in children [45]. Similar to food aid and supplementary feeding interventions in Indonesia, a program which provided food for children in exchange for work by parents found that while the intervention freed up household resources, there was a limited effect in terms of reducing child anemia [28].

Nutrition policies were also found to be important in addressing child nutrition challenges in crisis situations. In China, government responses to economic crises resulted in stable levels of nutritional status in children under 5 years [17]. Similarly, a study comparing the effect of child nutrition policy in Ethiopia found impressive improvements in child nutrition, including decreased stunting, wasting, and underweight prevalence [25].

### 3.3. Guidelines for Child Nutrition in Crisis Situations

The study reviewed seven documents of guidelines for child nutrition in crisis from various organizations, most of them United Nations agencies such as the WHO, UNICEF, and Food and Agriculture Organization (FAO), as well as from specialized agencies such as the IFE Core Group for infant and young child feeding in emergencies; Sphere Association, a group of humanitarian professionals; and the European Commission Directorate-General for Humanitarian Aid and Civil Protection. For more details, please see Appendix A.

The WHO’s “*Guiding principles for feeding infants and young children during emergencies*” [47] from 2004 is a fundamental guiding document indicating eight areas concerning feeding infants and young children during emergencies, namely: (1) breastfeeding, (2) breast-milk substitutes, (3) complementary feeding, (4) caring for caregivers, (5) protecting children, (6) malnutrition, (7) the acute phase of emergencies and assessment, and (8) intervention and monitoring. In 2005, the FAO developed the *Resource guide: Protecting and promoting good nutrition in crisis and recovery*. This guideline highlights the importance of food insecurity affecting infants, young children, pregnant women, and lactating women during crises, especially on matters of complementary feeding [48].

The IFE Core Group, a collaboration of UN agencies and NGOs, evaluated existing guidance, training materials, and resources related to complementary feeding in emergencies, and published *Complementary Feeding of Infants and Young Children in Emergencies* in 2009 [49]. This review document highlights the need for psychosocial support and supplemental feeding during emergencies. In 2017, the IFE core group developed *The Operational Guidance on Infant and Young Child Feeding in Emergencies* [50], which is mainly for operational actions, policy setting, and coordination, providing a comprehensive recommendation of policy and coordination needed during crises.

*The Infant and Young Children Feeding Emergencies: Guidance for Programming* of 2014 [51] is based on three international standard recommendations and guidance, namely: (1) *Infant and Young Children Feeding in Emergencies: Operational Guidance* (IFE 2007), (2) *Sphere Standards 2011*, and (3) the *International Code of Marketing of Breast-Milk Substitutes* (WHO 1981 and subsequent resolutions). This document recommends what must be integrated into services and assessment procedures in emergencies.

Lastly, in 2018, Sphere Association published “*The Sphere Handbook: Humanitarian Charter and Minimum Standards in Humanitarian Response*” [52]. This guidance suggests establishing supportive environments, managing malnutrition, assessment and monitoring procedures, and policy action during emergencies.

Guidance documents on feeding infants and young children during emergencies have become more inclusive over time. Recently, the WHO and UNICEF have developed *HIV and Infant feeding in emergencies: Operational Guidance as specific guidance for those living with HIV* [53].

This review found seven major concerns for guidance on young child and infant feeding in emergency: (1) breastfeeding, (2) breast-milk substitutes, (3) complementary feeding, (4) supportive environment, (5) assessment and monitoring, (6) management of malnutrition and (7) policy setting and coordination.

First of all, breastfeeding should not be affected by emergencies [47]. All efforts should allow infants to be exclusively breastfed up to 6 months of age and continue breastfeeding for 2 years or beyond. Mothers should be well-informed about the importance of exclusive breastfeeding for both the physical and psychological health of mother and child [52]. Breastfeeding may be affected by complex political, psychosocial, cultural, economic, and commercial influences and interactions [49]. There should be nutrition education interventions for all mothers, fathers, caregivers, and influential persons. There should be social/physical/psychological support mechanisms to address specific constraints [48] and all support should be integrated into services [51]. Furthermore, even mothers living with HIV (and whose infants are either HIV-infected or of unknown HIV status) should exclusively breastfeed their infants while being fully supported with antiretroviral therapy [53].

In an emergency, the use of breast-milk substitutes (BMS) should be strictly controlled; mothers should be well-informed and equipped to ensure safe preparation of BMS and the use of infant-feeding and artificial teats should be discouraged [47]. BMS packaging should properly provide information, with the instructions written in the local language with no brand name [48]. The implementation should align with the *International Code of Marketing of Breast-milk Substitutes.* “*The Infant and Young Children Feeding Emergencies: Guidance for Programming”* document recommends that [51]: (1) donated or subsidized supplies of breast-milk substitutes be avoided, (2) any decision to accept, procure, use or distribute infant formula in an emergency be made by informed, technical personnel in consultation with the coordinating agency and lead technical agencies, and governed by strict criteria; and (3) the use of bottles and teats in emergency contexts be avoided. Individuals should be assessed and consulted for the option of feeding only at the individual level. However, BMS feeding in minor groups should not interfere with protecting and promoting breastfeeding for the majority [47].

A recommendation for complementary feeding is mainly indicated in the *Protecting and Promoting Good Nutrition in Crisis and Recovery resource guide* [48]. It suggests that infants from 6 months onwards require nutritionally complementary foods to sustain their growth, development and health. Complementary foods should be prepared hygienically, easy to eat and digest, and locally made. Caregivers must secure uninterrupted access to ingredients and the basic needs of the household should be met.

Feeding infants and young children during emergencies requires an environment that supports mothers, infants, children, pregnant women and caregivers socially, physically and mentally. For a supportive physical environment, pregnant and lactating women should feel safe to use the facilities; a child-friendly space should encourage mothers to breastfeed their children up to 2 years or beyond [47,48]. Mothers and caregivers should be provided with the social and psychological support to cope with stress and to bring their children in for treatment. Some mothers may also need supported access to mental health services for perinatal depression [52], as well as peer support. Society should raise awareness of, and advocate, women’s rights. [48].

Assessment and monitoring of the impact of humanitarian action, young child feeding practices, and children’s nutritional status should be carried out to assess the needs and priorities of interventions and inform early decision-making. These recommendations are stated in *Infant and Young Child Feeding in Emergencies: Operational Guidance for Emergency Relief Staff and Programme Managers* [50] and *The Sphere Handbook: Humanitarian Charter and Minimum Standards in Humanitarian Response* [52]. What should be monitored are: (1) breastfeeding behavior (duration of exclusive breastfeeding, age at introducing of complementary foods, etc.), (2) food and nutrition assessments (food security, nutritional status); and (3) interventions (practices, policy, and code violations). Some surveys such as Multiple Indicator Cluster Surveys (MICS), Demographic Health Surveys (DHS), and child nutritional status can be used as pre-crisis data to evaluate the change and its impacts [50]. (See more details in Appendix A).

In times of emergency, children are prone to malnutrition [31]. The management of malnutrition is highlighted in The Sphere Handbook for humanitarian response. Key actions for management of malnutrition can be found in Supplement 2 and are stratified into three levels, i.e., management of moderate acute malnutrition, severe acute malnutrition, and micronutrient deficiencies. Dietary supplements may be necessary to meet nutrition requirements. Where fortified foods are not provided, IFE Core Group’s guidance suggested using micronutrients supplements differently in each specific group; children aged 6–59 months should be provided with multiple micronutrient supplements; and pregnant and lactating women should be provided with vitamin A and folic acid or multiple-micronutrient supplements [49].

Guidance on policy setting and operational action is prominently indicated in the *IFC Core Group Infant and Young Child Feeding in Emergencies: Operational Guidance for Emergency Relief Staff and Programme Managers* [50]. The intervention to protect, promote and support feeding infants and young children needs multisector efforts. The government or relevant agencies should develop policies and interventions that (1) protect, promote and support breastfeeding; (2) manage artificial feeding; (3) facilitate complementary feeding; (4) address the nutrition needs of pregnant and lactating women; (5) are compliant with the International Code of Marketing of Breastmilk Substitutes (BMS) and subsequent relevant World Health Assembly (WHA) Resolutions (the Code); (6) prevent and manage BMS donations; and (7) sustain proper infant feeding in the context of public health emergencies and infectious disease outbreaks. Additionally, the cultural context and the availability of services in affected places should be considered [49]. Leadership is required to coordinate operations from various sectors, and communication is essential to raise public and professional awareness [52]. Lastly, human capacity should be built by training programs and counselling mechanisms [52].

## 4. Discussion

Overall, this paper demonstrates the evidence of child nutrition and challenges in crisis settings. The evidence reveals that child nutrition is significantly affected by crises. Disasters, in particular, had impacts on children’s food intake, increasing malnourishment and anthropometric failures, which have a long-term impact on child development. Though the crisis per se may not be the main reason for malnutrition, it exacerbates existing livelihood problems among those in vulnerable settings. Several studies on the impacts of nutrition interventions in crisis settings found that interventions decrease pediatric mortality and morbidity in humanitarian crises; however, the effectiveness varies during droughts [18,35,37]. Cash-for-nutrition programs increase women’s knowledge and child feeding practices; however, there is no direct effect on child nutrition [41,42]. Education programs on breastfeeding have succeeded in significantly increasing breastfeeding practices, as compared to those who do not receive interventions [16,43]. Similarly, supplementary feeding and food aid interventions significantly impact nutrition outcomes and lessen anthropometric failures by reducing the prevalence of acute malnutrition and child underweight. This was observable by a higher linear growth as compared to those who do not receive interventions [24,44]. The critical feature of nutrition interventions was the community health volunteers who conduct the programs and are more familiar with the participants than implementers from outside. Nutrition policies are also essential in addressing child nutrition in crises. According to the guidelines, BMS should be strictly controlled in emergency settings. However, minor groups should not be interfered with, while breastfeeding in the majority should be promoted [39].

Feeding infants and young children during emergencies requires an environment that supports all people, including mothers, infants, children, pregnant women, and caregivers socially, physically, and mentally [39,40]. Multiple Indicator Cluster Surveys (MICS), and Demographic Health Surveys, and child nutritional status can be used as pre-crisis data to evaluate the change and its impacts [42].

The studies surveyed showed that nutritional status tends to worsen in all crises, both human-made and natural. Most papers mentioned human-made crises (as shown in the proportion of human-made paper vs. natural crises = 21 vs. 11). Most of the studies [21,22,27,44] on interventions were related to supplementary feeding programs that adopted the standard guidelines. Regarding community interventions, all interventions were implemented by local health volunteers, who were the main reason for the interventions’ success [41]. However, there is limited research discussing intervention mechanisms and the sustainability of programs [41].

The impact on nutrition status differs depending on the types of the nature of disasters [54]. For natural disasters, an immediate response to the crisis is required [55]. Stunting was found to be the most common indicator of malnutrition and is used as the nutritional index reported during crisis as a result of acute food shortages. [55]. As such, nutrition intervention programs such as supplementary and therapeutic feeding programs are very important, in particular among the most vulnerable groups since vulnerable populations may face greater difficulties in many aspects of life during the time of crisis than the well-off. These include financial barriers in accessing healthcare, lack of social support, and geographical distance that is oftentimes too far from the outreach assistance provided by the officials. These include pregnant women, lactating mothers, and severely malnourished children [56].

However, a humanitarian crisis and political emergencies may require alternatives rather than a single nutrition intervention. In humanitarian crises, cash transfers were reported as an effective intervention supporting vulnerable groups [57]. The study by Langendorf et al. suggested that a combination of cash transfer and food supplementary provided greater impact on severe acute malnutrition [58]. In fragile settings, evidence indicates that cash transfers may improve nutritional outcomes by increasing food supply and quality of food consumed [40]. More studies on health and nutrition education in emergency settings are needed to explore secondary interventions [40].

Given the complexity and unpredictability embedded in the nature of natural disasters and emergency crises, multimodal solutions are needed to address multifaceted nutritional challenges. Therefore, intervention effectiveness should be assessed based on the concept of complex intervention [40]. This is because the impact of these crises does not always appear in the form of the nutrition outcomes alone but the impact can mediate through all aspects of life, including financial welfare and family wellbeing.

Even though short-term programs are commonly prioritized when responding to crisis situations, negative consequences exist over time and therefore a long-term preventative is required to ensure program sustainability [35]. Local system capacity is critical to support community resilience, and engagement is necessary to ensure that emergency relief interventions can be delivered and target the right populations [59]. In this study, it is also important to consider the role of women in delivering nutrition intervention in the context of maternal and child health tasks [59].

Moreover, engagement with community leaders is mainly based on trust [60]. As such, community-led selection is an important mechanism to identify the right persons to understand those community contexts, needs and priorities [59]. Furthermore, integrating nutrition interventions during emergency situations into the national health service plan can strengthen the effectiveness and sustainability of the program implementation [59,61]. This does not mean that communities need to rely on themselves alone without adhering to the official health standards. Otherwise, it will cause other unexpected problems such as the increased use of homemade infant formula, which may be of low quality, and the misperception that breast-milk is not necessary in crisis times [62].

The study reveals some gaps. First, there should be more studies on nutrition during a human-made crisis, given that there are twice as many nutritional studies focused on natural disasters as there are on human-made crises. Second, the evidence on community specific interventions is sparse, since most intervention studies are about general nutritional education and supplementary feeding programs. Third, only a few studies focused on the sustainability of interventions; further studies should be conducted to analyze whether guidelines are implemented successfully. A primary study that employs a long-term follow up on the programs is recommended. This will help draw a lesson from the field in which the program sustainably works, by whom or by which knowhow, and in what conditions.

This study also has certain limitations. First, the studies were included based on the authors’ reading and judgements (based on the criteria which are mostly qualitative in nature), even though triangulation by a third party was carried out to reduce bias. Second, papers written in languages other than English were excluded. This may have resulted in selection bias by reducing the chance of retrieving papers from developing countries. Third, though this is rather the nature of scoping reviews rather than a limitation, it is worth noting that this review is just a precursor for further explorations for a more specific research question on various nutritional programs during the time of crisis [63]. Therefore, many findings above still lack quantitative evidence to support, oppose or pinpoint the effectiveness of a certain program (to answer this, one may need other research approaches, such as an empirical study or a meta-analysis) [64]. We may have missed papers from countries with severe crises (mostly from Africa or LMIC) as the papers from those countries are quite limited in number (compared to papers from high-income nations). Fourth, the method used in each paper greatly varies. The methods that directly analyzed the impact of the interventions (such as randomized trial or quasi-experimental design) were rarely used. Future research therefore should deliberately adopt a rigorous study design to assess the effectiveness and the impact of each nutrition-support intervention and determine which intervention works well in which context. Lastly, this study focuses on nutrition among children under the age of three in the context of natural or human-made crises and does not include the whole childhood period. Therefore, the findings should be interpreted with caution, given that there are other issues that need to be addressed for policy recommendations.

## 5. Conclusions

This study demonstrates that crisis situations adversely affect children under three years in terms of nutritional status, dietary intake, levels of anthropometric failure, and long-term development. Supplementary feeding and food aid interventions help to reduce nutritional and anthropometric failure. Educational interventions were successful in increasing the knowledge and practices of food intake but did not demonstrate any direct effects on nutritional status. Interestingly, all successful community-led interventions show that community health volunteers are potentially key driver of the success of the interventions. Further studies should be conducted to explore the sustainability of the crisis-related nutrition interventions, as well as the impacts of crisis-related nutrition guidelines. This study focuses only on nutrition among children under three years old and would benefit from extending its coverage to include children up to five years old. As the world’s children continue to face the risk of pandemics, environmental crises, and other disasters, both natural and human-wrought, the intersection between these crisis situations and nutrition will only be more vital.

## Figures and Tables

**Figure 1 ijerph-18-06637-f001:**
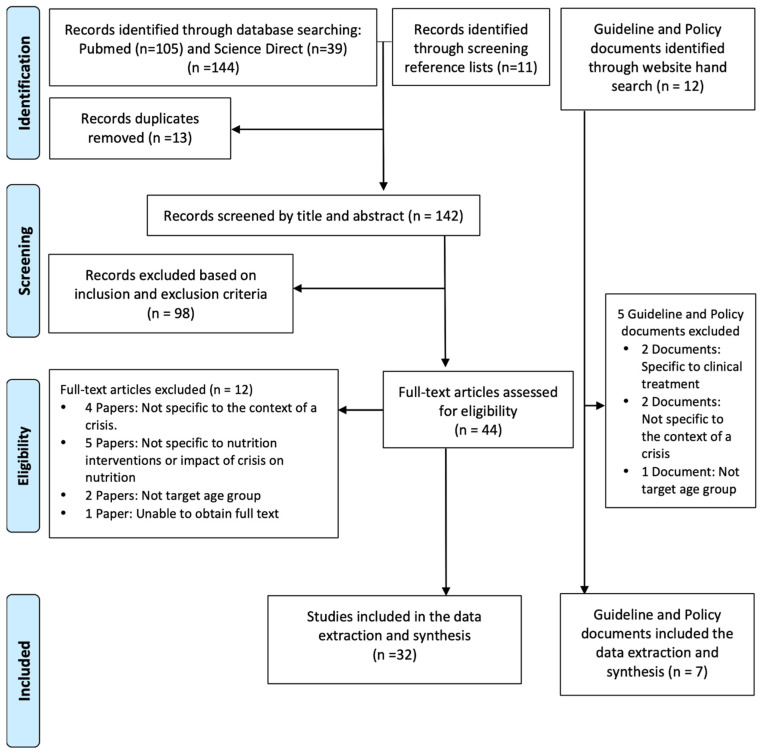
Article screening process.

**Table 1 ijerph-18-06637-t001:** Included studies.

Title	Author	Country/Location	Crisis	Aim	Target Group	Non-Intervention	Intervention	Assessment Method
Nutritional training in a humanitarian context: Evidence from a cluster randomized trial	Kurdi et al. (2020)	Yemen	Humanitarian	Reduce prevalence of child malnutrition through increasing women’s knowledge on child nutrition	Female relatives of Social Welfare Fund beneficiaries with children under 2 years or were pregnant	N/A	1. Cash for Nutrition 2. Nutrition education 3. Malnutrition screening	Cluster randomized trial, with infant and young child feeding knowledge and behavior outcomes and anthropometric measurements
Food-for-work programs in Indonesia had a limited effect on anemia	Moench-Pfanner et al. (2005)	Indonesia	Economic/Political	Free up household cash to permit increased consumption of micronutrient-rich foods to reduce the prevalence of iron deficiency anemia	1. Children (4–18 months old) 2. Mothers	N/A	1. Food for Work 2. Supplementary feeding program	Quasi-experimental design, measuring effect on income, Hb concentration, Vitamin A intake, and anthropometric measurements
Managing child malnutrition in a drought affected district of Rajasthan—A case study	Kumar et al. (2005)	India	Natural	Results from a case study of managing child malnutrition in drought affected areas	Children under 5 years	N/A	1. Supplementary feeding program. 2. Nutrition education	Prospective cohort
Fighting anaemia and malnutrition in Hebron (Palestine): Impact evaluation of a humanitarian project.	Magoni et al. (2008)	Palestine	Humanitarian	Evaluate impact of intervention	Children under 5 years	N/A	1. Supplementary feeding program. 2. Nutrition education	Pre-post evaluation
A cash-based intervention and the risk of acute malnutrition in children aged 6–59 months living in internally displaced persons camps in Mogadishu, Somalia: A non-randomised cluster trial	Grijalva-Eternod et al. (2018)	Somalia	Humanitarian			N/A	Cash for Nutrition	Non-randomized cluster trial
Children’s vulnerability to natural disasters: Evidence from natural experiments in Bangladesh	Ahsanuzzaman et al. (2020)	Bangladesh	Natural	Assess impact of cyclone on nutritional status of children under 5 years	Children under 5 years	Impact of crisis on nutrition	N/A	Analysis of data from the Bangladesh Demographic and Health Survey using a difference-indifference (DiD) estimation technique
Responding to health needs of women, children and adolescents within Syria during conflict: intervention coverage, challenges and adaptations	Akik et al. (2020)	Syria	Humanitarian	Document the provision and coverage of RMNCAH&N interventions	1. Children 2. Mothers 3. Adolescents	Implementation of interventions and barriers in crisis	N/A	1. Systematic/Document Review 2. Interview
Nutritional Change and Economic Crisis in an Urban Congolese Community	Cornu et al. (1995)	Congo	Economic/Political	Measure change in nutritional status of mothers and children under 6 years	1. Children under 6 years 2. Mothers	1. Impact of crisis on nutrition. 2. Implementation of interventions and barriers in crisis	N/A	Anthropometric assessment using cross-sectional survey data
Growth and anaemia among infants and young children for two years after the Wenchuan earthquake	Dong et al. (2014)	China	Natural	Monitor malnutrition morbidity and anemia prevalence in children aged 5–23 months	Children under 2 years	1. Impact of crisis on nutrition. 2. Implementation of interventions and barriers in crisis	N/A	Anthropometric measurements and hemoglobin concentration
Impact of disasters on child stunting in Nepal	Gaire et al. (2016)	Nepal	Natural	Assess association between disaster and stunting in children aged 5–59 months	Children under 5 years	Impact of crisis on nutrition	N/A	Assessment of data from Nepal Demography Health Survey (NDHS) 2011
Decreased attendance at routine health activities mediates deterioration in nutritional status of young African children under worsening socioeconomic conditions	Martin-Prével et al. (2001)	Congo	Economic/Political	Examine the role of routine health activity attendance and changes in nutritional situation among children aged 4–23 months	Children under 2 years	1. Impact of crisis on nutrition. 2. Implementation of interventions and barriers in crisis	N/A	Questionnaire-based cross-sectional study with anthropometric measurement
Nutritional status of Haitian children, 1978–1995: deleterious consequences of political instability and international sanctions	Mulder-Sibanda (1998)	Haiti	Economic/Political	Identify nutritional status in children aged 6–59 months	Children under 5 years	Impact of crisis on nutrition	N/A	Anthropometric assessment using cross-sectional survey data
Maternal and Child Health of Internally DisplacedPersons in Ukraine: A Qualitative Study	Nidzvetska et al. (2017)	Ukraine	Humanitarian	Explore perceived health, barriers to access to healthcare, caringpractices, food security, and overall financial situation of mothers and young children under 2 years	Children under 2 years	Impact of crisis on nutrition	N/A	Semi-structured in-depthinterviews
Household and community socioeconomic and environmental determinants of child nutritional status in Cameroon	Pongou et al. (2006)	Cameroon	Economic/Political	Assess factors associated with nutritional status in children under 3 years	Children under 3 years	Impact of crisis on nutrition	N/A	Anthropometric measurements using Demographic and Health Survey data
Impact of the 2008 economic and financial crisis on child health: a systematic review	Rajmil et al. (2014)	Global	Economic/Political	Provide an overview of the impacts of crisis on the health of children under 18 years	Children and Adolescents under 18 years	Impact of crisis on nutrition	N/A	Systematic/Document Review
Community volunteers can improve breastfeeding among children under six months of age in the Democratic Republic of Congo crisis	Balaluka et al. (2012)	Congo	Economic/Political	Raise mothers’ awareness of the benefits of breastfeeding and the need to practice exclusive breastfeeding from birth for a period of six months.	Pregnant women in their third trimester	N/A	Nutrition education	1. Anthropometric measurements 2. Survey data
Barriers to Breastfeeding in Disasters in the Context of Iran	MirMohamadaliIe et al. (2019)	Iran	Natural	Explore the barriers to appropriate lactation after disasters	Midwives	Implementation of interventions and barriers in crisis	Nutrition Education	Interview
Effectiveness of a Large -Scale Health and Nutritional Education Program on Anemia in Children Younger Than 5 Years in Shifang, a Heavily Damaged Area of Wenchuan Earthquake.	Yang et al. (2015)	China	Natural	Explored an ideal way to prevent anemia among children younger than 5 years after disasters	Children under 5 years	N/A	Nutrition education	Pre-post evaluation
Nutritional Status of Children during and post Global Economic Crisis in China	Chen et al. (2011)	China	Economic/Political	Identify changes in nutritional status in children under 5 years	Children under 5 years	1. Impact of crisis on nutrition. 2. Impact of intervention in crisis.	Nutrition policy	Anthropometric assessment using National Food and Nutrition Surveillance System (NFNSS) data
Examining the changing profile of undernutrition in the context of food price rises and greater inequality	Nandy et al. (2016)	Ethiopia and Nigeria	Economic/Political	Evaluate impact of food price increase on undernutrition in children under 5 years	Children under 5 years	1. Impact of crisis on nutrition. 2. Impact of intervention in crisis.	Nutrition policy	Analysis of cross-sectional demographic and health data with anthropometric measurements and food prices from the Food and Agriculture Organization’s (FAO) Food Price Index.
Impact of nutrition interventions on pediatric mortality and nutrition outcomes in humanitarian emergencies: A systematic review	Balhara et al. (2017)	Global	Humanitarian	Identify and describe the effect of nutrition interventions in disaster settings for children under 18 years	Children and Adolescents under 18 years	Impact of intervention in crisis	Overall	Systematic/Document Review
Drought exposure as a risk factor for child undernutrition in low- and middle-income countries: A systematic review and assessment of empirical evidence	Belesova et al. (2019)	Low- and middle-income countries	Natural	Assess drought as a risk factor for undernutrition in children <5 years of age	Children under 5 years	1. Impact of crisis on nutrition. 2. Impact of intervention in crisis.	Overall	Systematic/Document Review
Effects of biosocial variables on changes in nutritional status of rural Bangladeshi children, pre- and post-monsoon flooding	Choudhury et al. (1993)	Bangladesh	Natural	Determine the effects of biosocial variables on changes in nutritional status of children under 2 affected by flood	Children under 2 years	1. Impact of crisis on nutrition. 2. Impact of intervention in crisis 3.Implementation of interventions and barriers in crisis	Supplementary feeding program	1. Anthropometric measurements 2. Interview
Prospective Study on the effectiveness of Complementary Food Supplements on Improving Status of Elder Infants and Young Children in the Areas Affected by Wenchuan Earthquake	Dong et al. (2013)	China	Natural	Assess effectiveness of complementary food supplements	Children 6–18 months of age	N/A	Supplementary feeding program	Prospective cohort
Protecting child nutritional status in the aftermath of a financial crisis: Evidence from Indonesia	Giles et al. (2014)	Indonesia	Economic/Political	Protecting child nutritional status following financial crisis	Children under 5 years		Supplementary feeding program	Analysis of data from the Indonesia Family Life Survey
Relationship between food aid and acute malnutrition following an earthquake.	Hossain et al. (2009)	Pakistan	Natural	Assess relationship between food aid and acute malnutrition among children	Children under 5 years	N/A	Supplementary feeding program	Cross-sectional
Community -based management of severe and moderate acute malnutrition during emergencies in Sri Lanka: Challenges of implementation.	Jayatissa et al. (2012)	Sri Lanka	Humanitarian	Assess the impact of community-based management of acute malnutrition among children	Children under 5 years	Implementation of interventions and barriers in crisis	Supplementary feeding program	Prospective cohort
Spread fortified with vitamins and minerals induces catch -up growth and eradicates severe anemia in stunted refugee children aged 3–6 y.	Lopriore et al. (2004)	Algeria	Humanitarian	Assess the effect of a highly nutrient-dense spread fortified with vitamins and minerals in correcting retarded linear growth and in reducing anemia in stunted refugee children	Children under 6 years	N/A	Supplementary feeding program	Randomized Control Trial
Relationship of the availability of micronutrient powder with iron status and hemoglobin among women and children in the Kakuma Refugee Camp, Kenya	Ndemwa et al. (2011)	Kenya	Humanitarian	Evaluate the effect of the availability of home fortification with a micronutrient powder	Children under 5 years	N/A	Supplementary feeding program	Prospective cohort
Malnourished children and supplementary feeding during the war emergency in Guinea-Bissau in 1998–1999	Nielsen et al. (2004)	Guinea-Bissau	Humanitarian	Evaluate the effect of a supplementary feeding programs on malnourished children	Children under 5 years	Impact of crisis on nutrition	Supplementary feeding program	Prospective cohort
Treading water: The long-term impact of the 1998 flood on nutrition in Bangladesh	Ninno et al. (2005)	Bangladesh	Natural	Assess long-term impact of flood on nutrition of children under 5 years and impact of interventions	Children under 5 years	1. Impact of crisis on nutrition. 2. Impact of intervention in crisis.	Supplementary feeding program	Anthropometric measurements from household survey
The humanitarian emergency in Burundi: evaluation of the operational strategy for management of nutritional crisis	Rossi et al. (2008)	Burundi	Humanitarian	Evaluate the impact and appropriateness of programs for the management and treatment of severe malnutrition in emergency situations	Children under 5 years	N/A	Supplementary feeding program	Retrospective cohort

## Data Availability

The database of this study can be made available upon request by emailing the corresponding author.

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
