# Peer review of "Evidence on Child Nutrition Recommendations and Challenges in Crisis Settings: A Scoping Review"

_ijerph, 2021, doi:10.3390/ijerph18126637_

Round 1

Reviewer 1 Report

Review report: Evidence on child nutrition recommendations and challenges in crisis settings: a scoping review

The paper reviews existing evidence on nutrition response for early child development during crises. Based on the 32 studies, the results indicate that the existing evidence demonstrates the negative impacts of crises on nutritional status, diet intake, anthropometric failure, and long-term development.

Generally, this topic is interesting, and the structure of the paper is clear.  However, there are some problems need to be solved for the publication.

  1. For the reviews, the data processing is essential. reviews designed to summarize the large volumes of information are frequently published. When a review is done systematically, following certain criteria, and the results are pooled and analyzed quantitatively, it is called a meta-analysis. A well-designed meta-analysis can provide valuable information for researchers, policy makers. However, there are many critical caveats in performing and interpreting them. There are some critical issues need to be addressed in a meta-analysis, such as identification and selection of studies, heterogeneity of results, availability of information, analysis of the data. However, the data in present paper is not well structured, especially the information on publication bias is not clear enough. More information on the studies is necessary.
  2. It is highly recommended the authors to reference some relevant paper such as “Nelson, J. and P. Kennedy (2009). "The Use (and Abuse) of Meta-Analysis in Environmental and Natural Resource Economics: An Assessment." Environmental & Resource Economics 42(3): 345-377” and improve the review work.
  3. One well designed review would clear illustrate the main results and differences in present literature and the limitation of the studies including the method, data, experiment etc., The authors need to improve the analysis in the results part, and clearly reveal the main conclusions.
  4. Proofreading needs to be done before publication.

Reviewer 2 Report

Abstract

Line 22

“Good child nutrition......” the word ‘good’ is relative, could you consider to use either adequate or optimal...

Line 26

Consider change to, “In total, data from 32 studies were extracted and included in the data-extraction form”.

“long-term development” consider to be direct and perhaps say child development or long-term child development or something of the like.

Background

Line 36

General comment on the background

 Based on the purpose of the study, there are limited citations of literature from other countries that experience disasters and crises as a result of natural and man-made causes for example Ethiopia, Somalia, Nepal, South Sudan, Congo, etcetera. Considering the premise of providing global evidence, there is a need for varied citation for the literature synthesized.

Line 44

“In low-income and developing countries...” consider using the widely accepted

Line 48

“Low and middle-income countries” Consider being consistent with LMICS regarding comment on line 8.

Line 53

Besides...............social economic status” consider to rephrase the sentence and be more direct. For example, A World Health Organization (WHO) bulletin [6] reported that more than a quarter (27.9%) of children(age) are stunted, which affected their physical health, cognitive development, and socio-economic status.

Line 55

“According to academic and policy literature, inadequate dietary....” Consider to be more direct and begin “Inadequate dietary.................”

Line 58 and Line 44

Are disasters synonymous with crises for this paper? Consider the proper definition of the two terms and indicate the relation.

Line 63

“For example, In the Philippines ............” provide a citation for this example.

Line 65

“The programs .......and improving food intake” Consider adding a reference for the Philippine Plan of Action for Nutrition

Line 70

“Despite these data....is not routinely mentioned in academic literature.” Perhaps consider not generalizing this and keeping it specific to the Philippines.

Line 71

“Vulnerable populations ......disease.” Consider adding a citation for this statement.

Methods

Research question

Line 88

“The concept of this study is to Identify......” grammar.

Line 90

“Our final research question was....” How many research questions did you have? If you had many, could you consider writing them in the paragraph well indicated such as: ‘We wanted to determine 1) ...............2) ................................ and 3) ....................................?” Rephrase writing under the research question to be concise and clear.

Line 90

“Our final research question…….” Could you consider adding the perspective of the ‘study population’ in the question?  For example, around the world or globally

Line 93

Research question punctuation “?” Revise punctuation throughout the entire manuscript

Line 93

“We define the term “crisis” based on the World Health Organization...” This should have been defined or reflected a little earlier within the background since the term “crisis” is founded within the research topic itself.

Line 93

“based on the World Health Organization” .... also abbreviate in parenthesis if will be used further in the document for example in Line 115

Line 97

Which primary research question is the subsidiary questions directed to? Is the primary research question what you referred to as the final research question in line 53?

Line 110

Define UNICEF at the first time of use.

Line 119

“…...children under three… “ years? or age of three

Table 1. Included studies.

study 4

“Mangoni et al…” Include year of publication

Line 79

Consider adding a section describing how child nutrition was assessed. If various methods were used to assess for example dietary practices, and child growth; what were the more popular methods used in child nutrition assessment.

Results

Line 159

“…with child diet and nutritional status strongly affected, including the prevalence of malnutrition……” further clarity required on the specificity on the child dietary practices. Were they optimal or suboptimal? And how was the nutritional status affected? What was the change in the prevalence of malnutrition reported in this study? Was it higher, increased, lower, decreased? Be more specific.

Line 160

“Nidzvetska et al …” you may consider adding the corresponding citation number immediately after the author is being used to begin a sentence.

Line 160

The sentence also not clear, not sure if it’s tense in “Nidzvetska et al. ……….in children’s diet”. Did the research mean that displacement led to changes in the children’s diet? If that was the meaning, could you be more specific and report what exactly the study mentioned about the change? For example, Nidzvetska et al. (2017) reported inadequate dietary intake for children due to ration cuts and opting for alternative foods in cereals that were deemed cheaper. (Table 2, Nidzvetska et al. (2017))

Line 161

“Similarly, Mulder-Sibanda……………... a political crisis” refer to an earlier comment in line 160 on citation and specificity on reports outputs

Line 163

“Rajmil et al. ……………………………..” refer to an earlier comment in line 160 on citation and specificity on reports outputs. For example, what adverse effects? Did they report that they were harmful? Inadequate or unfavorable?

Line 163

Rajmil et al. …… and suggested that 280,000 – 50,000……..” Please rephrase to be concise and clear. Did they report an actual figure or estimate? You may consider replacing the word “suggested”

Line 167

“Choudhury et al. reported …” refer to an earlier comment in line 160 on citations.

Line 168

….; similarly, Dong et al. reported …” refer to an earlier comment in line 160 on citations.

Line 169

“….earthquakes.[23,25]” Need revising of the citation concerning punctuation (period).

Line 176

“Mulder-Sibanda reported that...........” refer to an earlier comment in line 160 on citations.

Line 179

“Three studies found........; four found.............” consider citing immediately after three studies [ ] ... four [ ]............

Line 179

Consider revising grammar for plural. Consideration of results from many studies

Line 187

“In Bangladesh ..... and being underweight” Consider adding the odds ratios and confidence intervals of the likelihood of stunting and underweight.

Also, consider adding a justification for the results. Answer the “why” or mention why the authors of the studies reviewed found a likelihood for an increase in stunting and underweight rates.

Line 189

“......moderate and severe stunt ....” grammar. Also be consistent with nutrition status description throughout the manuscript

Line 190

“.....an odd ratio of 0.42 and 0,59..” correct odd ratio figure punctuation to conform to other figures in the manuscript

Line 190

“In Cameroon ..................underweight” rephrase the grammar of the sentence for a better understanding. Also, consider specificity to children in rephrasing. Also, consider adding the odds or other probability indicators used by the original authors of the article reviewed.

Line 192

“Natural disasters were ......nutrition” provide the rationale for this statement.

Line 193

“Dong et al. reported .....” refer to an earlier comment in line 160 on citations. Also, revise sentence grammar

Line 194

“Ninno et al. reported ....” refer to an earlier comment in line 160 on citations.

Line 195

“...long-term growth is unable...” grammar
Line 196

“...similarly, Ahsanuzzaman et al. reported....” refer to an earlier comment in line 160 on citations.

Line198

“Major factors....include....” grammar.

Line 199

“Choudhury et al. reported ...” refer to an earlier comment in line 160 on citations.

Line 201

“. while Dong et al. reported ...” grammar and refer to an earlier comment in line 160 on citations.

Line 202

“...a higher percentage of anemia among girls than boys ....” indicate the figures of the percentage in parentheses.

Line 202

“Gaire et al. ....” refer to an earlier comment in line 160 on citations.

Line 204

“Pongou et al....” refer to an earlier comment in line 160 on citations.

Line 205

“....economic crises have more effects on children...” be specific about what effects; Is it on growth, feeding practices, development, or something else?

Line 218

“ Balhara et al. reported that..................................” refer to an earlier comment in line 160 on citations.

Line 220

“...found that varying effectiveness of nutrition interventions during droughts vary, and interventions...” clarify the meaning of the statement. Consider using not using “vary” two times in a sentence.

Line 222

“The review identified studies that report a decline in child underweight levels following interventions, as well as studied with limited evidence of mitigating the impact of interventions.” Consider reviewing the tenses. Also, cite at least two studies because of the plural component of the statement. Preferably, cite the studies immediately after the word “studies.”

Line 225

Considering removing the sentence “Of the two cash-for-nutrition programmes...” It sets up a comparison which is negated by the use of “both” after the sentence.

Line 226

“Kurdi et al. reported...” refer to an earlier comment in line 160 on citations.

Line 235

“Of the 6 nutrition education programs, 5 included or focused on breastfeeding” Consider rephrasing, for example; ‘Of the 6 studies based on nutrition education programs, the majority (5) focused on ....’

Line 236

“According to MirMohamadaliIe et al., training...” refer to an earlier comment in line 160 on citations.

Line 249

“....higher infant weight at 12 months (8.42kg compared to non-intervention group at 7.95kg) [35], and...” Are these mean weights? If so, please consider being specific.

Line 251

“Kurdi et al. reported.....” refer to an earlier comment in line 160 on citations.

Line 251

Tenses, grammar

Line 252

“.........resulted in significant improvements to nutrition outcomes and anthropometric failures......” Consider mention the specific nutrition outcomes. Also, use “reduce” or “lower” anthropometric failures.

Line 261

“....significant changes from extreme stunting to stunting....” be consistent with the description of stunting categories i.e., moderate, severe. Also see the comment at Line 539

Line 384

In times of emergency, children are prone to malnutrition...” provide citations for the statement.

Conclusion

Line 520

Grammar

Line 290 -291

UNICEF description should have come much earlier in the manuscript

Line 410

Provide some citations

Line 412

“……children’s diet and food intake…” what is the difference between diet and food intake?

Line 412

“….affect children’s diet…….” Be concise about the type pf “affect” that happens on dietary practices

Line 414

I am not sure that you can say with such affirmation from your literature synthesis. You could consider using the words “may not be…”

Line 418 – 420

Be specific on the context of “child nutrition” that you refer to twice in the sentence

Line 421

Breastfeeding practices or breastfeeding? Is this is consistent with the lingual used in the IYCF-E guidelines?

Line 424

Grammar

Line 426

Grammar

Line 440 – 441

Citations required

Line 442

Rephrase “lack of papers…” to perhaps “limited research”. You may have conducted a good search but you can’t be conclusive that the search methods based on only the keywords you used to exhaust all research that may exist on the topic or related. Also, you did exclude all studies that were not conducted in English.

You may also consider adding this as a recommendation for future researchers rather than a critique of academia which may also not be entirely true.

Line 443

“Note ……...” This statement is out of context, maybe rephrased or removed from the manuscript.

Line 447

“For natural……” qualify this statement with a citation. Does this sentence mean that manmade crises/disasters may not require an immediate response?

Line 447

“Also stunting …shortage” you may need to discuss this statement more because stunting (Length/height for age) is widely known as an indicator of chronic malnutrition rather than what you described

Line 447 – 452

Also include a rationale for such programs to the “vulnerable groups” what the programs exactly do to these people to address malnutrition.

Line 484

I think the first recommendation is circumstantial unless you want to discuss how many manmade crises exist that haven’t had research conducted in them compared to the natural crises. Consider rephrasing or remove

Line 490

“Studies …..conducted” This is a bit ambiguous because we expect scope reviews, systematic analyses, and metanalyses to provide such information. See line 504 -509, this seems like something you highlighted. Consider revising.

Line 492

Grammar

Line 502

Grammar

Line 502

“The…..adverse…..” revise wording to be concise

Need for being concise “…..effects on children under three….” Refer to comment in Line 119

Line 509

“............................ determine the success of the standard guidelines and the sustainability of the interventions.” You need to rephrase the sentence for clearer context, based on the findings of your review, there seems to be a lot of ‘success’ from the existing guidelines. Are you wanting to mean that more research to be done to explore more benefits of the guidelines?

Supplementary 1

Revise title format

Crosscheck grammar and punctuation

Font consistency

Refer to ijerph table font guidelines for large tables(https://www.mdpi.com/journal/ijerph/instructions#figures)

Supplementary 2

Revise title format

Crosscheck grammar and punctuation

Font consistency

Refer to ijerph table font guidelines for large tables(https://www.mdpi.com/journal/ijerph/instructions#figures)

Reviewer 3 Report

Nice and timely review.

Consider discussing rise of homemade infant formula during times of disaster. 

Round 2

Reviewer 1 Report

The authors revised  the paper, but the big concern is still not clear enough. All those researches were carried out based on extremely different conditions. It is necessary to take the data, methods, and other information into consideration when doing review works, no matter it is a systematic or scoping review. otherwise its implications are limited.
